# The Role of Local Leaders in the Implementation of Adult-Initiated Motor Skill Development and Physical Activity in Preschool—A Case Study

**DOI:** 10.3390/ijerph182413119

**Published:** 2021-12-12

**Authors:** Trine Top Klein-Wengel, Jonas Vestergaard Nielsen, Søren Smedegaard, Thomas Skovgaard

**Affiliations:** 1Research Unit for Active Living, Department of Sports Science and Clinical Biomechanics, University of Southern Denmark, 5230 Odense, Denmark; twengel@health.sdu.dk (T.T.K.-W.); tskovgaard@health.sdu.dk (T.S.); 2Research and Implementation Centre for Human Movement and Learning, University of Southern Denmark, 5230 Odense, Denmark; sosm@ucl.dk; 3University College Lillebaelt, UCL Campus Odense, Niels Bohrs Allé 1, 5230 Odense, Denmark

**Keywords:** preschool, motor skills, physical activity, preschool leaders, qualitative research, implementation

## Abstract

Good motor skills (MS) and physical activity (PA) are considered important for children’s physical, social, and psychological development. The Motor skills in PreSchool (MiPS) study-Denmark aimed to optimize children’s MS through weekly PA sessions. The aim of this paper is to explore the role of local preschool leaders and their strategies in influencing the implementation of MiPS into daily practice. Leaders from all seven preschools in the project were interviewed. The results show that preschool leaders used communication (setting an agenda and dedicating speaking time to address the program at staff meetings) and reflective questioning about the pedagogic staff’s current practice in relation to the program (adding focus on MS and PA) as their main course of action. Through this form of communication and reflective questioning, the preschool leaders aimed to involve the staff and heighten their sensemaking in the existing practice while also ensuring that the implementation of the program was kept in progress. In sum, future implementation of an MS and PA initiative in preschools should put emphasis on a shared responsibility between leaders and staff combined with an adaptive approach in which the existing practice is reshaped rather than just increasing the workload of the pedagogic staff.

## 1. Introduction

Having good motor skills (MS) and engaging in physical activity (PA) are considered to be important for children’s physical, social, and psychological development [1,2,3]. Early life MS development is particularly important in order to create a foundation for more complex movement activities in daily living and sports later in life [1,3]. Recommendations for PA put forward by the World Health Organization also state that children (1–5 years old) should get at least 180 min of PA a day, of which one hour should be of moderate-to-vigorous intensity [4]. Globally, however, only a small proportion of children meet these recommendations [5,6]. This has highlighted the need for effective implementation strategies to increase PA levels in the primary years of childhood [7,8,9]. It is unknown to which degree Danish 1-to-5-year-old children comply with the guidelines, yet about eight percent of Danish children were reported to have MS difficulties upon reaching school age [10]. Thus, a need has been raised to strengthen the focus on MS and PA development in Danish preschools [11].

Approximately 84% of 3-to-5-year-old children living in developed countries receive out-of-home care [12] and spend an average of 30 h a week in formal childcare [13]. This is also the case in Denmark as the majority of preschoolers spend 6–8 h in childcare [14]. Furthermore, the preschool setting provides opportunities for reaching and engaging children from different socioeconomic levels and different residential areas (e.g., city and countryside) [15,16,17]. Thus, the preschool setting could be a prime setting for interventions to promote MS and PA in early childhood [2,8,11].

Intervention studies in preschools have been shown to increase children’s PA [18,19,20]. Still, activity levels are, to a large extent, influenced by the specific activities in individual preschools [21], and there appears to be great variation in the amount and form of the PA incorporated in individual preschools [17]. This is the case despite an existing focus on PA through a specific pedagogical curriculum framework for all Danish preschools [22]. Disseminating behavior-related initiatives into a real-world context such as preschools can be challenging; there is a need for in-depth evaluations with greater attention to local contexts and practical implications of the initiatives [23,24,25]. This entails a better understanding of the consistency and internal validity of the MS and PA initiatives by, among other things, documenting whether and why (not) they were implemented. Such a focus on the implementation process heightens the transferability by allowing practitioners and decision makers to identify and adopt promising initiatives that fit their local context. Thus, there is a clear need for further knowledge to provide in-depth perspectives from real-world initiatives to support decisions on how to implement PA in the preschool setting.

### 1.1. MiPS

In order to add to the knowledge regarding the implementation of MS and PA interventions in preschool, the present study explores the Motor skills in PreSchool (MiPS) study-Denmark. MiPS is a program set in the “real-world”—meaning that the settings, populations, and conditions are relatively uncontrolled and “normal”. The program was initiated in seven preschools located in Svendborg Municipality, Denmark, in January 2017 and has been running for four years [15]. The intervention was planned to run all year across all seasons. The program was developed via a collaboration between representatives from the research team, the participating preschools, the municipality, and other experts in order to enhance the contextual fit and sense of ownership among all stakeholders. The aim of the project was to optimize children’s MS through the training of both coordination and balance, gross and fine motor challenges, as well as the vestibular, tactile, and kinesthetic senses [15]. To obtain this, four main elements were established: (i) each week the pedagogic staff should arrange at least four PA sessions focusing on MS development and lasting no less than 45 min; (ii) the PA sessions were to support the training of both coordination and balance, gross and fine motor challenges; (iii) the pedagogic staff should align the PA sessions with current pedagogic goals; and (iv) children should engage in daily PA with high intensity. To secure the quality of the program and support the implementation process, all pedagogic staff should participate in five competence development courses (37 h distributed across a two-month period). The aim of the course was to deliver tailored knowledge, skills, and capacity to the staff in order to facilitate the implementation of the program [15]. Due to the real-world design of the study the local preschools, in collaboration with Svendborg Municipality, had the responsibility to carry out the intervention; thus, researchers mostly became engaged in the evaluation (effects and process) rather than in the implementation and upholding of program fidelity. This evaluation included an RCT with seven of the municipal preschools participating in the intervention and nine preschools acting as controls. The participating preschools were randomized, stratified for socioeconomic background. A mean socioeconomic index for each preschool was developed based on family type, education, and income, and this was dichotomized to above or below the median for all the included kindergartens [15]. Of the seven participating preschools three were located in rural areas, two in suburban areas, and two in urban areas. The current study adds a focus on the seven preschools undergoing the intervention and their implementation processes.

### 1.2. The Importance of Local Leaders

Local leaders (in this case within childcare institutions) are important operators when implementing health promotion and prevention programs in childhood settings such as MiPS [26,27,28,29]. In their ecological framework, Durlak and DuPre emphasize more generally how local leaders can impact implementation by specifying tasks, establishing consensus, and managing the overall process of a given program that is to be implemented [30]. Local leaders also influence implementation processes and outcomes through their daily management, coaching, and encouragement of employees [26,30].

In a study specifically examining the implementation of an initiative aimed at enhancing the learning and development of preschool children, the commitment of the local leader was found to be imperative for the program to be introduced and implemented [29]. The study also examined barriers for implementation and found that lack of commitment from the local leader (e.g., not participating in program meetings or missing out on discussions and decisions related to program progress) was one of them [29]. In a Danish context, a recent report from Nielsen et al. highlighted the important role of local preschool leaders if new daily practices were to work properly in varying contextual settings [31]. Among other things, the report concludes that local leaders have to involve the staff and help facilitate a general culture of ongoing reflection and day-to-day evaluation in order for a program to become a new and sustained normal in the preschool [31]. Although the importance of local childcare leaders has been highlighted in relation to the general quality of practice, there is, to our knowledge, only a scarce number of studies examining the role of local leaders in implementing new daily routines in preschool. Furthermore, there is an emerging need for research aimed at identifying effective strategies and factors influencing implementation processes in a real-world setting [32,33]. MiPS has the potential to gain far-reaching perspectives because the initiative is applied at seven institutions, all with different physical conditions and cultures. Svendborg Municipality is also comparable to the rest of Denmark in terms of age distribution, gender, and income [15]. The effectiveness of MiPS will be determined through a cluster randomized controlled trial (RCT) that focuses on the improvement in MS as well as several secondary effects. However, there is also a need for evaluations with greater attention to the local preschool context and practical implementation strategies used in MiPS. Thus, both quantitative and qualitative implementation measures will be explored in order to support the overall evaluation of MiPS. The current study will contribute to the evaluation by focusing on the role of local leaders through qualitative investigation.

Hereby, the aim of the current study is to explore the role of local preschool leaders and their strategies in influencing the implementation of MiPS into daily practice. These insights and perspectives from local preschool leaders and their strategies to implement MiPS can potentially accommodate the continuing need for strategies on how to translate and disseminate MS and PA initiatives into everyday preschool practice [31,34,35].

### 1.3. Theoretical Framework

This study uses both Karl Weick’s theory of sensemaking [36] and Jody Hoffer Gittell’s theory of relational coordination [37] as the governing frameworks. The frameworks are used as starting points for the collection of empirical data and in the data analyses to investigate the role of local leaders in the implementation of new practices concerning MS and PA in Danish preschool settings.

Sensemaking, a concept introduced by Weick in the 1970s, has been described as the process by which we turn often many-sided contexts and conditions into a “situation that is comprehended explicitly in words and that serves as a springboard into action” (Weick et al., 2005, p. 409). Such types of sensemaking are often highlighted as a core leadership competence for the complex and changeable working environments that characterize large parts of both private and public sector activities [38]. Building on this perspective, implementation leadership is very much about building a sense of meaningfulness and capacity to act in relation to the new, and thereby provide a sense of positive progress [39]. In accordance with the aim of this study, and building on the approach initially coined by Weick, sensemaking is the starting point for examining the ways in which local preschool leaders strive to make all employees see how MiPS makes sense for them and their work efforts, thereby influencing the implementation process in preschools.

Jody Hoffer Gittell’s theory of relational coordination offers a useful theoretical framework that can be used to describe the impact of coordinated communication and cooperation in processes that require changes in organizational design and/or key work processes [37]. Importantly, this theory emphasizes two important dimensions for strong performances and (inter) organizational collaboration: communication and relations. Communication, Gittell states, needs to be frequent, timely, precise, and problem-solving while good relations are characterized by shared objectives, knowledge sharing, and mutual respect [37]. The theory of relation coordination will be used to examine to what degree and in what ways local leaders communicate and build relations with the pedagogic staff in order to generate shared goals and knowledge conducive to securing program implementation.

## 2. Materials and Methods

The scientific theoretical foundation of this study is based on the philosophical theory of pragmatism. Through a pragmatic approach, attention is directed at actions that can facilitate potential outcomes deemed relevant and valuable in local practice and exploring how such actions can be implemented in wide-spread practice [40]. These pragmatic perspectives have a clear link to implementation science, and the focus on the application of innovations in real-world conditions as well as the promotion of transparency enable practitioners and policymakers to integrate evidence into practice [24,35,41].

This study uses semi-structured interviews to explore the role of local leaders in the implementation of MiPS. Interviews were done with local leaders at all seven preschools in MiPS. The interviews investigated the leaders’ perceived influence on the implementation of MiPS. This included (i) their role and critical skills required for implementation, (ii) how they, as leaders, had facilitated and/or hindered the implementation, and (iii) how the existing preschool culture and norms influenced their ability to implement MiPS. All questions were inspired by a theoretical framework containing Karl Weick’s theory of sensemaking [36] and Jody Hoffer Gittell’s theory of relational coordination [37]. An example of the interview guide and questions can be found in Appendix A. Prior to the interviews, the interviewees received an information letter stating the aim of the investigation, the role of their involvement, and the estimated duration of the interview. The information letter did not include specific interview questions.

### 2.1. Interviews

A list of the current preschool leaders at the seven preschools participating in MiPS was created in collaboration with the municipal project managers. All seven leaders were invited to participate in an interview, and all agreed.

Local leaders were interviewed about their role, considerations, and actions in relation to the implementation and maintenance of the program. The leaders were interviewed using a semi-structured interview guide. All participants were female and they had between 5 and 22 years of experience as leaders. Four of the participants had been involved as local preschool leaders since the initiation of the program in 2017. Among the three remaining participants, one had moved from a leader position at another preschool in the municipality that also participated in the program; one had moved from a leader position at another preschool in the municipality that did not participate in the program; and one had moved from a leader position at a preschool in another municipality.

### 2.2. Conducting the Interviews

Interviews were conducted during October and November 2019. The duration of the interviews was 40–60 min. All interviews were carried out by a postdoc with previous interview experience as well as experience in the preschool area. The interviewer had no personal or professional affiliation to the interviewees. The interviews took place in a quiet room at the worksite/preschool of the local leaders in order to make the situation less formal and more on the participants’ terms [42]. Furthermore, the interviewer used the first few minutes of the interview to establish a more relaxed setting by asking various preliminary questions about the interviewee’s background and story as a preschool leader.

The interviews were carried out using a semi-structured interview guide and included open-ended questions. Semi-structured interviews were used in order to ensure a structure regarding the theoretical framework while also allowing the interviewer to diverge from the interview guide to follow interesting elements that arose during the interview and enable the interview to be driven by the interviewee’s experiences [43]. To heighten the credibility of the information retrieved during the interview, the interviewer frequently tried to summarize the points made by the interviewee or asked the interviewee to summarize their points themselves. This helped validate and minimize potential misunderstandings of the information given.

### 2.3. Analysis Strategy

Interviews were analyzed using qualitative content analysis [44]. Ultimately, this entailed a systematic description of the empirical data through coding [44]. The interviews were initially transcribed and the produced transcripts were then pseudonymized. Subsequently, the first author thoroughly familiarized herself with the data prior to coding the material. This familiarization was done by repeatedly reading the material while taking notes. Both the first and the second author independently trial-coded a large portion of the interviews according to a coding frame. The coding frame was constructed to embrace a set of categories representing the main themes of the theoretical framework applied in the study: a (i) relational part and a (ii) communicative part. Subsequently, the first and the second author also performed an independent data-driven trial coding of which subcategories inductively emerged, describing underlying themes and elaborating on how the main categories related to the leaders influenced the implementation and maintenance of the program. This inductive process was continued until no new themes were observed in the data and saturation was achieved [45]. These subcategories were used to adjust and refine the coding framework. Through a peer debriefing process, the first and second author systematically and critically examined the consistency and adequacy of their individual coding [46], followed by a discussion in order to reach consensus. Subsequently, all data that had been used in the trial coding were recoded with the adjusted coding frame along with the rest of the interview data by the first author. An example of the coding frame can be found in Appendix B.

This analysis allowed for an in-depth understanding of the local leaders’ influence on the implementation process, their ability to integrate the program into daily practice, as well as the adaptations made to the program in order to fit the local organizational structures, resources, and facilities. Data were analyzed using Nvivo 12 (QSR International Inc., Burlington, MA, USA).

### 2.4. Ethics

Written informed consent, also containing consent for publication, was obtained from the participants prior to the interviews. All interviews were audio-recorded and performed in private rooms at the workplaces of the interviewees. Further, the interviewer emphasized that the participants were free to withdraw from the interview at any time and without explanation. This study was approved by the Regional Committees on Health Research Ethics for Southern Denmark (S-2015-0178) as well as by the Danish Data Protection Agency (2015-57-0008).

## 3. Results

The results report on the dominant themes that emerged from the analysis relating to the local leaders’ role and strategies for implementing MiPS into daily practice, namely: (i) frequent preschool leader communication and staff involvement served as key elements to facilitate the implementation process; (ii) leaders were able to help build a sense of meaningfulness and relevance to the existing preschool practice; and (iii) preschool leaders recognized that implementation is an ongoing process, recognizing the need to ensure sustained attention on the project elements in the preschools.

### 3.1. Communication and Involvement as Key Elements

All local preschool leaders highlighted the importance of frequently communicating about MiPS with the staff. This entailed communication during both meetings and everyday life, involving the staff in the implementation of the project. This was viewed by leaders as one of the key elements to facilitate the implementation process. The analysis shows that leaders actively used frequent communication with and the involvement of the staff to attain what Gittel describes as shared knowledge, shared goals, and mutual respect [37]. All are factors that they found important in different ways in their effort to realize the successful implementation of MiPS. The focus on shared knowledge, shared goals, and mutual respect allowed the pedagogic staff to adapt and be specific with how their practice should evolve.

In my experience, attaining the knowledge… the “why” and the “how to” [through the professional development course]… it won’t stick unless I made sure to create a framework around it… to make sure that there was time to reflect on the knowledge… and so I asked the pedagogic staff “what does it mean to you”… to start a common and shared reflection… What does this mean to you, and what does this mean for us as an institution?(Leader at preschool three).

#### 3.1.1. Frequent and Accurate Communication

All local preschool leaders highlighted that frequent communication was an important instrument to assist the implementation of MiPS. Through regular and up-front communication about the project (implications for daily practice and the staff’s end goals concerning their work with the children), the local leaders aimed to support a common understanding of the project. This was mainly done through discussions of MiPS during staff meetings and everyday practice situations. The extent to which the leaders prioritized time for this varied. Five of the local preschool leaders expressed that they prioritized dedicated time for project communication both during meetings and in everyday situations. One local leader mainly prioritized project communication during meetings, and the last local leader mainly used everyday practice situations to inform and discuss the project with the pedagogic staff. To a large extent, communication about MiPS during staff meetings (both the entire staff group and smaller groups) aimed to achieve shared knowledge and shared goals. Six of the seven leaders stated that they continually encourage the staff to reflect on the project and try to maintain a sense of cohesion and accurate understanding of the project to make sure they are working toward the same goals when implementing and adapting project elements that reform their daily practice. One of the preschool leaders expressed this, as follows:

We have created a culture with focus on the daily dialogue. We communicate to each other if something doesn’t work, if there is a need for a small adjustment and so on… Often they [staff] come to me to reflect and get my opinion: This … just didn’t work, how do you think we can change it? … So, that is a part of our joint and daily reflections…(Leader at preschool six).

#### 3.1.2. Problem-Solving Communication

All seven participants mentioned a need to take on the role as “the leader” and take charge when the implementation of MiPS was challenging, e.g., when the staff found the program difficult or confusing, or when they lacked focus regarding the integration of MS and PA into their daily practice. In such situations, the local preschool leaders expressed that communication was their main approach—both through directing attention to the problem as well as awarding time to discuss solutions. When asked about their handling of challenges during the implementation of MiPS, two of the local preschool leaders mentioned:

So I try to simplify…, what is the most important thing we need to do right now, and then help create time to do that…(Leader at preschool two).

When there is a challenge, we need to point it out and talk about it and what it is that makes it so difficult and how we can solve it together as a team. The thing I can do as a leader is to say: “Is this something I should try to fix, or is it something you are able to change yourselves?”(Leader at preschool six).

#### 3.1.3. Involvement of Staff

Although the local preschool leaders emphasized that they had to uphold a rather noticeable amount of control and make final decisions on key issues, they pointed out the importance of involving the staff in order to find the most optimal way to continue the implementation of MiPS into daily routines.

All the local preschool leaders expressed that involvement of the staff was an important factor for the implementation process. The staff were involved at the project’s initiation and were able to express their perspectives on and impact the implementation of MiPS in their local preschool. The local preschool leaders emphasized that they were very keen on continuing this involvement in order to establish shared responsibility for the project and to promote a sense of meaningfulness in their implementation of MiPS. The creation of a reflective environment was highlighted by six of the local preschool leaders as an important approach to involve the staff in the implementation process. In their communication with the staff, the local preschool leaders focused on having a curious approach. Through this, preschool leaders wanted the staff to reflect upon their existing practice and how they might rethink or innovate it in a meaningful way in accordance with the MiPS program’s focus on MS and PA. The goal was to support them in their individual adoption of the program. Several of the local preschool leaders highlighted that they encouraged such reflections as a way to find a balance between bottom-up and top-down approaches—involving staff in how they could change their pedagogic practice and daily work routines while the local preschool leaders set a direction for the work and had the overall responsibility for implementing MiPS.

I have the overall responsibility. I must ensure that there is progress. But I don’t do it alone, I have the staff…, and during the everyday work with the children and implementing that focus on movement and motor skills, they are the ones [the staff] who are responsible…(Leader at preschool one).

As the local leader I outline some main goals, but they [the staff] also help set those goals to ensure that they become co-owners of it. If I am solely dictating and say, “I have specified these four things we are going to implement”, and then they [the staff] are the ones who are going to carry them out, then it is simply not going to be effective…(Leader at preschool four).

### 3.2. Building a Sense of Meaningfulness and Relevance

#### 3.2.1. The Importance of Adaptability

In Denmark, preschools are required to work with a specific pedagogical curriculum framework. Several of the local preschool leaders emphasized that elements from MiPS have been integrated into the existing work with the curricula and thereby their daily pedagogic practice. A majority of the local preschool leaders express that the project fits in with the existing curriculum, which eased the implementation of MiPS:

I think what is important is that MiPS has become a part of our consciousness… Instead of “just” implementing the project, MiPS has been integrated into our existing curriculum goals…, in that way movement has become something we just do all the time…(Leader at preschool five).

In general, the local preschool leaders expressed that they were able to maintain a lot of the existing pedagogic practice in the preschool while also implementing MiPS. This was partially because the project was deemed well-fitted to the preschools’ existing focus, practice, and facilities. The local preschool leaders are convinced that this harmony between the “old” and the “new” had a positive effect on the implementation of MiPS. Part of the reason for this is that the staff appreciated the opportunity to develop their existing practice, which they already found meaningful, instead of building up and creating meaning in a completely new practice. Furthermore, most of the local preschool leaders expressed that the project goals were broadly formulated, which made them easier to understand and adapt into their local setting. Still, the local preschool leaders participating in the study expressed that it required a great deal of time to evaluate their current practice and how project elements could be adapted in a meaningful way for the staff as well as the children. The leaders prioritized time to discuss the project with the staff, especially in the beginning of the implementation period. This made it possible for both leaders as well as staff to contemplate how MiPS could be implemented in a way that would make sense in their local context. The local preschool leaders were convinced that this possible adaptation of the program strengthened their ability to implement MiPS. The following quote illustrates the local preschool leaders’ experiences in relation to adapting the project:

It has been important to implement this in the simplest and easiest way possible … meaning that it has not been the aim to build something completely new but to develop our existing practice…(Leader at preschool two).

#### 3.2.2. The Importance of Committed Leaders

Five of the leaders found the project relevant and were committed from the start of the project. They found the focus on MS and PA both important and meaningful as a pedagogic approach as well as in the promotion of child development. According to the leaders, they used their existing commitment to argue for the project’s relevance, benefits, and outcomes in their daily preschool practice to motivate the staff. In addition, all preschool leaders prioritized time for the staff to attend professional development courses which provided them with core knowledge and a common starting point. One of the leaders expressed:

I find that it [MiPS] is extremely relevant! It is a project that has made sense to me, also in relation to what we stand for in our local preschool… So, it has been easy and valuable for me to implement it…(Leader at preschool two).

One of the preschool leaders, however, did not give the project high priority due to other pedagogical and professional concerns in their local preschool. Still, the leader expresses that she found the project relatively easy to adapt to the existing practice, and even though MiPS did not have a high priority, elements from the project were implemented into daily practice:

At our preschool, some think… myself included… that it is more important to focus on social skills and how the children are actively engaging as a group [rather that have a direct focus on MS and PA]… Then, again, some of the methods we use when working with… for example, with social skills… it may well be adopted from the MiPS project…(Leader at preschool one).

### 3.3. Implementation as an Ongoing Process

Six of the local preschool leaders expressed the need to keep an ongoing focus on MiPS to secure long-term maintenance of the project elements in their daily practice. All of the local preschool leaders believed that it has been and still will be important to keep the staff focused on MS and PA activities and keep their daily pedagogic work related to the project. Leaders express that this maintenance perspective is attained by continuous focus from their side and through initiating dialogues with the staff regarding their local pedagogic practice in relation to MiPS:

It certainly requires that I retain a focus on the project [MiPS] and that movement is still a foundation for our pedagogic work… while I am fully aware that there are other things in the curriculum that we also have to live up to…(Leader at preschool seven).

Still, some of the leaders expressed that maintaining a focus on the elements of MiPS could become a challenge. This is due to natural changes that always happen in preschools, e.g., in the diversity among the group of children or new requirements from the political level that must be met.

Most of the local preschool leaders expressed that they use daily communication to support an ongoing dialogue and reflection on the activities and everyday practice related to the project. In order to ensure that the focus on MS and PA is sustained as part of their preschools’ practice, leaders emphasized the importance of continuing this assistance of the staff by giving them time to reflect on and plan their pedagogic practice and help them maintain a sense of meaningfulness.

## 4. Discussion

The success of behavior-related preschool programs is heavily dependent on effective implementation. The results show that preschool leaders used communication (setting an agenda and dedicating speaking time to address the program in connection with staff meetings) and reflective questioning to probe the staff’s current practice in relation to the program (adding focus on MS and PA) as their main course of action. Through this form of communication and reflective questioning, preschool leaders aimed to involve the staff. In addition, the leaders recognized implementation as an ongoing process, and through the commitment of the staff they strained to ensure that the program was maintained. In the following sections, these results are discussed further in relation to the literature. These insights and perspectives from local preschool leaders and their strategies to implement MiPS may potentially accommodate the continuing need for strategies on how to translate and disseminate MS and PA initiatives into everyday preschool practice [31,34,35].

### 4.1. Implementing a New Focus on MS and PA in Preschools

Implementing programs such as MiPS in real-world settings with already-established and complex everyday practices is typically rather challenging [47,48]. Often, implementation is centered around a high degree of fidelity and the action of merging a pre-defined “something” into practice [49,50]. Yet, one often-recurring challenge is that the program that needs to be implemented is seen as an “add-on” to an already busy workday [51,52]. The term “add-on” is often described as initiatives that are applied on top of existing responsibilities and tasks to be done—leading to an increased workload for the staff and local leaders [53]. In contrast, the term “add-in” has been used to represent initiatives intended to become part of already-established activities or as a resource to supplement existing tasks [53]. In the current study, local leaders actively involved the staff through daily communication and reflective questioning in order to promote their innovation and integration of MiPS into daily practice and routines. Such add-in approaches increase local involvement through co-creation and the bottom-up involvement of frontline staff, serving to engage them in integrating new approaches and didactical perspectives [35]. In the current study we found that this entailed leaders to find a balance between (i) adapting the program to their existing practice and (ii) rethinking the existing practice.

#### 4.1.1. Adapting the Program to the Existing Practice

An add-in approach is argued to be a more accessible approach for the implementation of health promotion components into childcare institutions [53]. As seen in the results of the current study, this is connected with the ability of the program to reflect feelings of relevance and meaningfulness. Furthermore, fitting a program to the local implementation sites has been highlighted as an important factor in ensuring commitment, implementation, and maintenance [30,48]. As an example, an add-in approach was used in the development and implementation of the SHAPES study, applying a highly flexible program that was adapted over a 3-year period by involving preschool staff [54]. Through this 3-year development, it was highlighted that the staff’s feelings of ownership of the project and the opportunity to influence it in a direction that they found meaningful had a positive influence on their participation in and commitment to PA [54]. In the SHAPES study, the add-in process was promoted and organized by a research team. However, the current study indicates that this process can be supported by committed leaders when dealing with real-world programs that do not have the same close collaboration with researchers. Still, despite the attempt to engage with and involve professional stakeholders, a new program is likely to face implementation challenges when workload and new tasks are added without additional resources (e.g., time, knowledge, facilities, or additional staff) [48,55,56]. Results from the current study indicate that this entails that local leaders need to find ways of rethinking their existing preschool practice within the new MS and PA perspective.

#### 4.1.2. Rethinking the Existing Practice

Whether an add-on or add-in approach is used, an aspect is added that demands, at the very least, a rethinking of the collective work situation of key stakeholders in order to enable and facilitate stringent implementation. In line with the add-in approach, preschool leaders and staff were involved in the development of MiPS while leaders also report that the program was fairly flexible and that it was possible to adapt it to their local settings. However, results of the current study show that the local leaders did not merely focus on adding new tasks and workload but also on identifying to which degree MS development and PA already existed: how they could rethink their existing practice within this focus. This highlights the fact that local leaders, among other things, should take responsibility for clarifying the prioritization of staff responsibilities and tasks—including what should be given a lower priority, removed, or reconsidered when new initiatives and tasks are to be introduced. Additionally, emphasis should be set on how new programs can support and qualify the existing practice rather than adding new, incoherent tasks. In the present study, this is mainly displayed by the preschool leaders’ efforts during the workday to facilitate what the staff actively consider to be meaningful ways to integrate MS and PA activities into their existing pedagogic practice. However, this requires special attention from local leaders to focus on frequent communication and reflective questioning in order to help the staff to realize what aspects of their daily practice they should continue doing and what should be transformed or merely removed. In combination, leaders devoted time to staff meetings to follow up on the reflections and ideas of the staff. Leaders incorporating such systematic feedback processes have been found to ensure that adaptations to the original program are reasonable and sustained over time [54]. In sum, future programs implementing an MS and PA initiative in preschools might gain by emphasizing the building of a shared responsibility among leaders and staff, combined with an adaptive approach in which the existing practice is reshaped rather than additional workload just being added.

### 4.2. Local Leaders as Vital Implementation Support

The results of the current study indicate that the dedication of local preschool leaders played an important role in the continuous support and management of the implementation of MiPS. Staff groups are often viewed as key implementers as they are the ones implementing a given program in the existing practice or supplementing the existing practice with added procedures [30,54]. Still, the commitment and motivation of preschool leaders have been noted by others to make them important agents in order to reach successful implementation as they are generally the ones who initially adopt a new program as well as provide vital and ongoing support to the staff [26,29]. The preschool leaders in the current study did, however, emphasize the importance of the pedagogic staff as the main program providers and seek to promote their active involvement in integrating MS and PA as key components of daily activities in preschools. Thus, leaders also had a core focus on assisting the staff in developing or rethinking their existing pedagogic practice. This was mainly done through dialogue aiming to cause the staff to reflect upon how their current practice was consistent with the desired practice (focus on MS and PA). Most leaders express that this assisting dialogue with the staff helped secure a contextual fit as well as the pedagogic staff’s adoption of the program. Besides motivation and competences [56,57], the literature concerning organizationally based implementation often emphasizes the importance of broad and meticulous stakeholder involvement in order to increase the probability of adoption and create real and sustainable change [30,49,54]. Such involvement resembles what Lewin et al. characterize as democratic leadership (as opposed to authoritarian and delegative leadership), in which leaders participate actively in the implementation process by guiding the staff and encouraging inputs [58]. Such leadership is characterized as highly effective [58] and was also deemed to be an effective approach to aid the maintenance and overall quality of the new practice involving MS and PA. Thus, we recommend that preschool leaders engage in such reflective dialogue in order to involve and encourage their staff in the implementation of new MS and PA initiatives.

### 4.3. Maintaining a Focus on MS and PA

The results of the current study highlight local leaders’ ability to support implementation as a major capacity for change. However, programs that are able to achieve implementation success still need to avert program drift or deterioration over time [35]. The results indicate that preschool leaders have been aware of the need for ongoing implementation focus during the program’s 3-year lifespan. One of their strategies to uphold its implementation was through codependent teamwork with the pedagogic staff. This codependent teamwork between the local leader and pedagogic staff entailed a top-down leadership (ensuring an ongoing focus on MS and PA) and a bottom-up involvement (engaging staff to reflect upon their own practice and find solutions that would make sense in their daily work routines). Applying such a combination of top-down and bottom-up teamwork has been highlighted as an important factor to achieve successful implementation and increase the probability of maintenance [35]. Furthermore, preschool leaders have to a large extent worked towards sensemaking [36] through ongoing communication as a key element in their implementation approach.

It should also be recognized that not all of the preschool leaders prioritized the project; thus, they did not engage in the same degree of ongoing communication and reflective questioning. This was due to other pedagogical and professional concerns. Still, these leaders express that core project elements were implemented into daily practice due to a relatively easy fit into their existing practice. This indicates that a contextual fit is an important supplement to the ongoing staff involvement and sensemaking process—that the involvement of the staff in planning the program during the pre-implementation stages is a key element aiding the implementation [30,48,54]. Thus, although local leaders can facilitate the implementation of MS and PA into local practice, they still benefit from support on a structural level (e.g., politicians and municipal project managers) in terms of enabling early stage involvement, for instance [35,59].

### 4.4. Strengths and Limitations

The importance of local leaders when implementing health initiatives in an educational setting has been established throughout the literature [26,27,29,30,31]. Still, to our knowledge there is only a scarce number of studies examining the role of local leaders when implementing new daily routines in preschools. One of the strengths of this study is the added attention to the importance of leadership in implementation as well as the literature exploring the implementation of health promotion initiatives—specifically warranted in a Danish preschool context. Furthermore, there is a call for evaluations of programs in a real-world setting [32,33] in addition to greater attention to be paid to local contexts and the practical implications on how to assist implementation and broader maintenance processes [23,24,25]. Still, we recognize that this study has limitations. Firstly, only four of the seven participating leaders had been involved as local preschool leaders since the initiation of the program in 2017 and therefore had the opportunity to influence the implementation process from the beginning. Still, it should be noted that all participants had been local leaders at their present preschools for some time and had considerable experience as preschool leaders. Regarding the interviews, these were performed two years after MiPS was initiated, enabling the possibility for some degree of recall bias of the information dating back to program initiation. Generally, it would have been preferred to follow and document the implementation process from the initiation of the program in 2017, unfolding the various implementation stages that the program has undergone.

Secondly, both the perspectives of the staff [54,60,61,62] and the end users (children and their parents) [63,64] are vital in order to obtain successful implementation. However, these perspectives have not been explored. In particular, insights on strategies used to introduce and secure the backing of the staff are missing. Furthermore, preschool leaders mention political agendas and financial support as an uncertainty for future practice. This structural level, including political and economic perspectives, has not been explored, yet could contain vital information on how to secure the implementation and long-term maintenance of preschool programs [35,59]. Thus, future research should explore end users, staff, and the structural level in order to provide additional insights and reveal if relevant implementation factors can be identified in the preschool setting [20,54,60,61].

Thirdly, the transparency and applicability of the knowledge produced in the study should also be mentioned. The extensive focus on local leaders and implementation strategies as well as contextual descriptions of MiPS has provided a unique insight into a real-world innovation containing an additional focus on MS and PA at Danish preschools. Although the results regarding the leader’s strategies and influence on the implementation were discussed and mainly supported by international literature, it should still be highlighted that the empirical foundation originated in a Danish preschool context. The analysis of leadership factors influencing the implementation of MiPS should have strengthened their transferability and assisted preschool leaders and decision makers to translate the results of the study. In a Danish context, Svendborg Municipality is close to the average Danish municipality on a number of relevant aspects [15]. Still, a majority of the preschools in the study found the added focus on MS and PA relevant and in alignment with existing values and priorities. This could challenge the possible transferability to other preschools not having a grounded interest in MS and PA.

Lastly, future research would benefit from gaining a more comprehensive understanding of fidelity levels and program outcomes. Based on the study, it is not possible to estimate fidelity nor if health outcomes were achieved. Translation of MiPS could have been strengthened by determining fidelity and, if any measured health outcomes had occurred, allowing decision makers to identify the relevance of the program and must-have elements when adapting the program to their individual context [34,35].

## 5. Conclusions

The current study explored the role of local preschool leaders and their strategies to influence the implementation of an MS and PA intervention in a Danish preschool setting. The study highlights the role of local preschool leaders as an important component in achieving successful implementation. A take-home message is that the implementation of MS and PA can be supported by local preschool leaders’ frequent communication and reflective questioning about the pedagogic staff’s current practice in relation to the project. The early involvement of preschool leaders and staff in the development of MiPS in combination with the leaders perceiving the program as fairly flexible strengthen the ability to adapt MiPS to their existing preschool culture and norms. In relation to this, another take-home message is that leaders should not merely focus on adding new tasks but also on identifying how the “new” already exists or what it should replace. Through a form of reflective questioning, preschool leaders both applied top-down and bottom-up approaches in their implementation strategy, which may have heightened the staff’s sensemaking and the sustainability of the new focus on MS and PA. In sum, the future implementation of an MS and PA initiative in preschools should put emphasis on a shared responsibility from both leaders and staff, combined with an adaptive approach in which the existing practice is reshaped rather than additional workload just being added.

## Data Availability

The data presented in this study can become available on request from the corresponding author. The data are not publicly available due to legal and privacy issues.

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
