# Peer review of "The Role of Local Leaders in the Implementation of Adult-Initiated Motor Skill Development and Physical Activity in Preschool—A Case Study"

_ijerph, 2021, doi:10.3390/ijerph182413119_

Round 1
Reviewer 1 Report
Thank you for conducting this study, and I hope these comments help improve the paper.
Introduction:
Could authors explain why the qualitative study needs or clearly explicate the rationale of the study?
Materials and Methods:
1. Can the authors provide the details of the demographic information of participants?
2. What is the approach used to select interviewees?
3. Codebook or coding information can be provided to let the readers know more about the information
4. Line 184-185: authors stated that all participants were female. Could authors elaborate more about the reason why only female interviewees?
Findings & Discussion:
As the authors stated that the purpose of the study is to explore the role of local leadership in the implementation of MiPS. However, I couldn't find the answers to what role of local leadership play? Can authors give more explanations for the critical research question?
In the current form, the new value and further significance need to be elaborated on in-depth. Moreover, could authors provide more take-home messages for practitioners?
Language checking:
English writing: the manuscript needs careful editing by someone with expertise in English editing paying particular attention to grammar, especially the sentence structure, so that the goals and findings of the study are clear and precise to the reader.
Author Response
Reviewer 1:
Thank you for conducting this study, and I hope these comments help improve the paper.
Authors’ comment: Thank you for your review and feedback. We have tried to respond to all your comments below.
Comments:
Introduction:
- Could authors explain why the qualitative study needs or clearly explicate the rationale of the study?
Authors’ comment: Please find that we have further described the MiPS study and how the qualitative part will support the overall evaluation design (Line 127-133).
Materials and Methods:
- Can the authors provide the details of the demographic information of participants?
Authors’ comment: As described in line 197 all leaders participating in the interviews were female with leadership experience ranging from 5-22 year (Line 198). Furthermore, we have added a short section regarding the sample of the seven preschools participating in MiPS and how they were sampled based on a randomization, stratified for socioeconomic background (Line 90-98). Also, of the seven participating preschools four were located in rural areas, two in suburban areas and two in urban areas (Line 90-98). Lastly, we would like to highlight that the municipality of Svendborg is comparable to the rest of Denmark in terms of age distribution, gender, and income (Line 124-126).
- What is the approach used to select interviewees?
Authors’ comment: We are not sure that we have understood your concern/comment correctly. We would like to direct your attention to line 192-193 stating that interview participants were recruited based on a list of the current preschool leaders at the seven preschools participating in MiPS. The list was created in collaboration with the municipal project managers.
- Codebook or coding information can be provided to let the readers know more about the information
Authors’ comment: Please find that we have provided a short Appendix B visualizing the coding frame used.
- Line 184-185: authors stated that all participants were female. Could authors elaborate more about the reason why only female interviewees?
Authors’ comment: We sought to interview the sitting leaders of the participating preschools to explore their role in the implementation of MiPS. It just so happened that all the leaders were female. In relation, we would like to highlight that both leaders and the pedaogic staff in Danish preschools is mainly occupied by women.
Findings & Discussion:
- As the authors stated that the purpose of the study is to explore the role of local leadership in the implementation of MiPS. However, I couldn't find the answers to what role of local leadership play? Can authors give more explanations for the critical research question?
Authors’ comment:
We are a bit unsure if we understood the comment correct. The aim of the study was to explore the role of local preschool leaders and their strategies to influence the implementation of MiPS into daily practice. We do not find this aim to be critical of the role of the local leader – we highly acknowledge that leaders have an impact on implementation of new initiatives in preschool (section 1.2). Based on this and our pragmatic approach (Line 172-175), attention is directed to actions that can facilitate potential outcomes deemed relevant and valuable in a local preschool practice (in this case the preschools in MiPS).
As stated during the discussion the local preschool leaders in MiPS had an important impact/role in the implementation of the MS and PA initiatives. This was mainly through strategies such as frequent communication and reflective questioning regarding the pedagogic staff’s current practice in relation to the project (securing contextual fit and ownership) as well as not merely focus on adding the new tasks but also on identifying how the ”new” already exists or what it should replace (ensuring a fit between resources and staff workload). These points were also highlighted in the conclusion.
- In the current form, the new value and further significance need to be elaborated on in-depth. Moreover, could authors provide more take-home messages for practitioners?
Authors’ comment: We have made some revisions and additions throughout the discussion and in conclusion. We hope that this is to a satisfactory degree.
Language checking:
- English writing: the manuscript needs careful editing by someone with expertise in English editing paying particular attention to grammar, especially the sentence structure, so that the goals and findings of the study are clear and precise to the reader.
Authors’ comment: We are sorry to hear that you did not find the use of English writing to be satisfying. We have previously used a professional editorial agency in order to secure the quality of the English writing and grammar in the manuscript. We have been through the manuscript making minor editing in the sentence structure. If you have any specific examples of sentences or grammar that need correction, we are happy to oblige – also, this would help us in the argument for a new and more thorough editing by the agency we have used.
Reviewer 2 Report
Thank you for addressing all of my previous concerns and comments. I believe this manuscript is much improved and will be well received by ijerph readers.
Author Response
Reviewer 2:
Thank you for addressing all of my previous concerns and comments. I believe this manuscript is much improved and will be well received by ijerph readers.
Authors’ comment: We are glad that you found that the manuscript had the sufficient quality to publish. We thank you for your previous comments, enabling the manuscript to gain the needed quality.
Reviewer 3 Report
The topic of your study is interesting
My recommendations are the following:
The Methods section does not appear from the abstract. There are no data on participants. I recommend introducing these aspects, in the order of the journal requirements. Present the results by purpose.
Strengths and limitations are far too extensive, I recommend being more focused.
In the bibliography at index 4 it appears twice a year, I recommend the correction.
In the bibliography you have editing errors, I recommend correction. Follow the rules of the journal.
Author Response
Reviewer 3 comments:
- The topic of your study is interesting. My recommendations are the following:
The Methods section does not appear from the abstract. There are no data on participants. I recommend introducing these aspects, in the order of the journal requirements. Present the results by purpose.
Authors’ comment: Thank you for noticing this. Please find that we have revised the manuscript and added a short sentence concerning the methods used and participants.
- Strengths and limitations are far too extensive, I recommend being more focused.
Authors’ comment: Thank you for this comment. We have tried to revise the strengths and limitation section, shortening it and making it more focused. However, we would like to highlight that parts of the section have been produced based on relevant comments/needs from another reviewer – e.g. the case of transperancy and transferfability. We hope the added revisions is satisfying so we can obligde to all the reviewers.
- In the bibliography at index 4 it appears twice a year, I recommend the correction.
Authors’ comment: Thank you for noticing this. Please find that this has been corrected.
- In the bibliography you have editing errors, I recommend correction. Follow the rules of the journal.
Authors’ comment: Thank you for this comment – please find that we have been through the bibliography with the rules of the journal in mind and have corrected a number of errors.
Round 2
Reviewer 1 Report
Thanks for the revisions and answers. One suggestion here, could authors consider providing a more detailed codebook/coding information to readers instead of a current short one. Specifically, Category, Definition, and Examples should be included in the codebook.
Reference: Mayring, P. (2004). Qualitative content analysis. A companion to qualitative research, 1(2), 159-176.
Author Response
Reviewer 1:
Thanks for the revisions and answers. One suggestion here, could authors consider providing a more detailed codebook/coding information to readers instead of a current short one. Specifically, Category, Definition, and Examples should be included in the codebook.
[Reference: Mayring, P. (2004). Qualitative content analysis. A companion to qualitative research, 1(2), 159-176.]
Authors’ comment: Once again thank you for your review and feedback – we believe that they have improved the manuscript. We acknowledge that the previously provided codebook was very short and could benefit from being elaborated. Please find that we have added two columns in the table in Appendix B: ‘definition’ and ‘examples’. Hereby each category is now label with the definition used during the analysis as well as a relevant example of this from the empirical material.
This manuscript is a resubmission of an earlier submission. The following is a list of the peer review reports and author responses from that submission.
Round 1
Reviewer 1 Report
Thank you for conducting this study, and I hope these comments are helpful for improving the paper.
1. Introduction
The study should outline its rationale, its objective, and how the research questions will be addressed in the introduction part. The authors need to make clear statements about how to bridge the previous research gaps in this manuscript. In this form, it is not clear, and there are few explanations of the rationale for the study. Additionally, in this section, there is a need to elaborate on the findings of previous studies (more detailed) and state the researchable points that you have found through the literature. More importantly, to establish a robust inner logic to illustrate the research questions. Some linkage can make the whole piece of work more coherent and less choppy. Overall, the introduction would be improved by clarifying what new information can be learned that cannot be covered in the previous literature.
Some other following contents need to be refined:
Line 31- define what age of children
Line 32-33: The authors stated that children accumulated insufficient physical activity globally, any evidence from the local context in Denmark? And how about motor skill proficiency? Authors could consider adding relevant literature to explain the status quo.
Line 103: what is the meaning of “selected” preschool settings?
2. Materials and Methods
2.1: what is the approach used to select interviewees?
2.2: conducting the interviews: how the interview questions were developed?
2.3 analysis strategy: specify the approach of data analysis in this section- deductive thematic analysis?
3. Results
Some findings are related to the role of leader and implementation strategies but lack detailed discussions on the other two key sections-facilitators and barriers to implementations and perceived culture change in the preschool. In this part, I think the authors must reanalyze the transcripts to make them clear to the readers and echo the previous argument in the manuscript (lines 133-134).
Any demographic information regarding the participants can be provided?
4. Discussion
What are the unique implications are expected from this study that moves beyond what we already know from previous literature? I suggest authors can add some related arguments accordingly to make theoretical and methodological contributions.
Besides, I have difficulties finding specific take-home messages for practitioners. In particular, there is a lack of discussions for the implementation of research on physical activity or motor skill intervention.
Others:
English writing: the manuscript needs careful editing by someone with expertise in
English editing paying particular attention to grammar, especially the sentence structure, so that the goals and findings of the study are clear and precise to the reader. Besides, some typo errors and grammar (tense) also need to be modified: such as line 331: leaders’; line 401: “… A high degree…” the capital A here needs to be changed; line 463: whished; line 395: should use the past tense to change “will be” into “were”. Also, there are some descriptions in the manuscript are ambiguous, such as line 371-372 “it could become a challenge to maintain focus on the elements from MiPS in the future” and line 464 “this helped secure a …” please specify “it” and “this” refers to what.
Author Response
Authors’ comment: Thank you for your thorough review of the manuscript. We acknowledge that the study had some limitations and highly appreciate your comments. We have strived to address each of your concerns below. We do find that it has helped improved the manuscript. All the revisions are marked in yellow in the revised manuscript.
Introduction:
- The study should outline its rationale, its objective, and how the research questions will be addressed in the introduction part. The authors need to make clear statements about how to bridge the previous research gaps in this manuscript. In this form, it is not clear, and there are few explanations of the rationale for the study. Additionally, in this section, there is a need to elaborate on the findings of previous studies (more detailed) and state the researchable points that you have found through the literature. More importantly, to establish a robust inner logic to illustrate the research questions. Some linkage can make the whole piece of work more coherent and less choppy. Overall, the introduction would be improved by clarifying what new information can be learned that cannot be covered in the previous literature.
Authors’ comment: Thank you for this comment. We have tried to address your concern throughout the introduction. As a part of our response, we have clarified the need for implementation research from behavior-related initiatives in a real-world context (such as MiPS). We have especially tried to highlight the rationale for the study in the text surrounding aim of the study (line 113-120 & 122-125).
- Some other following contents need to be refined: Line 31- define what age of children
Authors’ comment: Please find that we have clarified that it is 1–5-year-old children (line 35-36)
- Line 32-33: The authors stated that children accumulated insufficient physical activity globally, any evidence from the local context in Denmark? And how about motor skill proficiency? Authors could consider adding relevant literature to explain the status quo.
Authors’ comment: Thank you for highlighting this. It is highly relevant to address the evidence from a Danish context. Please find that we have revised and addressed this in line 37-42.
- Line 103: what is the meaning of “selected” preschool settings?
Authors’ comment: Thank you for noticing this. Please find that we have revised and now stating ‘a Danish preschool setting’ (line 131)
Materials and methods:
- What is the approach used to select interviewees?
Authors’ comment: Thank you for this relevant comment regarding the sampling of interviewees. Please find that we have added this information on line 180-182.
- Conducting the interviews: how the interview questions were developed?
Authors’ comment: As stated in line 172-173 “All questions were inspired by the theoretical framework containing Karl Weick’s theory of sensemaking (36) and Jody Hoffer Gittell’s theory of relational coordination (37).”
- Analysis strategy: specify the approach of data analysis in this section- deductive thematic analysis?
Authors’ comment: The description of a ‘deductive thematic analysis’ in the beginning of the results section was an unfortunate mistake. As described in section 2.3 Analysis strategy, the study applies a ‘qualitative content analysis’. We are very pleased that you highlighted this, so that we could correct the mistake.
Results:
- Some findings are related to the role of leader and implementation strategies but lack detailed discussions on the other two key sections-facilitators and barriers to implementations and perceived culture change in the preschool. In this part, I think the authors must reanalyze the transcripts to make them clear to the readers and echo the previous argument in the manuscript (lines 133-134).
Authors’ comment: We find that the findings are in line with the aim of the study: to explore the role of local preschool leaders and their strategies to influence the implementation of MiPS into daily practice. Still, as a response to your comment we have specified the purpose of the interviews (the key-sections we believe you were referring to) in line 169-171. This was an inadequate description, that did not clearly state the detailed and extensive focus on leadership. In addition, we would argue that the leaders (respondents) perceived i) role ii) actions/leadership have facilitated or hindered the implementation and iii) how the existing preschool culture and norms influenced their ability to implement MiPS, is represented and discussed in the manuscript.
Based on your comments we have, however, tried to make the flow and arguments clearer to the reader. This by specifying the text in the beginning of the results section, now clarifying how the themes from the analysis correlate with the aim of the study. Secondly, we have tried to specify how the results fulfil the aim in the conclusion section of the manuscript, also adding focus on the three mentioned key-sections from the purpose of the interviews (line 603-611).
- Any demographic information regarding the participants can be provided?
Authors’ comment: As mentioned in the text we have described the gender and range of experience of the leaders. Also, we have shortly referred to Svendborg municipality’s comparability to the rest of Denmark in terms of age distribution, gender, and income (line 117-118)
Discussion:
- What are the unique implications are expected from this study that moves beyond what we already know from previous literature? I suggest authors can add some related arguments accordingly to make theoretical and methodological contributions. Besides, I have difficulties finding specific take-home messages for practitioners. In particular, there is a lack of discussions for the implementation of research on physical activity or motor skill intervention.
Authors’ comment: We are not sure that we fully understand the comment regarding “a lack of discussions for the implementation of research on PA and MS intervention”. We have tried to strengthen the study’s link to the field of ‘implementation research’, as well as highlighted that MiPS is a so called ‘real-world’ program, entailing that it is the local preschool in collaboration with the municipality, that has the responsibility to carry out the intervention – thus researcher become mostly engaged in the evaluation rather than implementing and upholding the program.
In accordance with your other comments (and the comments from the other reviewers) we have tried to clarify the unique implications expected from this study through out the manuscripts.
Others:
- English writing: the manuscript needs careful editing by someone with expertise in
English editing paying particular attention to grammar, especially the sentence structure, so that the goals and findings of the study are clear and precise to the reader. Besides, some typo errors and grammar (tense) also need to be modified: such as line 331: leaders’; line 401: “… A high degree…” the capital A here needs to be changed; line 463: whished; line 395: should use the past tense to change “will be” into “were”. Also, there are some descriptions in the manuscript are ambiguous, such as line 371-372 “it could become a challenge to maintain focus on the elements from MiPS in the future” and line 464 “this helped secure a …” please specify “it” and “this” refers to what.
Authors’ comment: Please find that the language has been corrected throughout the manuscript. We hope that we have qualified the language to a satisfying degree.

Reviewer 2 Report
Overall, I believe this paper is well-written and very interesting. It provides evidence to support implementation strategies for PA and MS interventions in early childcare settings which is widely and critically needed. There are a few comments I would like to see addressed, however, in addition to a careful edit for grammar issues.
Introduction
- It is written in the last line of the first paragraph that, “Combined with the lack in conforming with PA guidelines, this highlights the need to implement strategies to reduce sedentary time in the primary years of childhood”. While I agree, I believe the evidence stated above related more to increasing MPVA rather than decreasing sedentary time - both of which are important but different goals
- Can you clarify whether MiPS is meant to run all year? Or is there a limited time during the school year that it runs?
Methods
- Please identify your research philosophy (also called a paradigm or philosophical position)
- I’d like to see more consistency with the label of the ‘local leaders’ – they have so far been referred to as ‘local leaders’ ‘acting leaders’ and ‘acting local leaders’
- We need to have more details about the interviewer: who conducted the interview? what was their relationship to the respondents (i.e., were they known to the interviewee? Would they have a reasonable level of trust with them to ensure full candor in their answers?) Might there be any biases related to this relationship? How experienced were the interviewer(s)?
- Please further describe the credibility of your data: was there any attempt to ensure that data was interpreted in a way that the interviewee intended? i.e. member validation either during or after the interview? Was data saturation reached when conducting your inductive analysis?
Discussion
- What degree of transference do you think exists in your data? How applicable might these results be to other Danish preschools? What about to an international population?
Author Response
Authors’ comment: Thank you for your thorough review of the manuscript. We find the comments both relevant and legitimate. We have strived to address each of your concerns below. All the revisions are marked in yellow in the revised manuscript.
Introduction:
- It is written in the last line of the first paragraph that, “Combined with the lack in conforming with PA guidelines, this highlights the need to implement strategies to reduce sedentary time in the primary years of childhood”. While I agree, I believe the evidence stated above related more to increasing MPVA rather than decreasing sedentary time - both of which are important but different goals
Authors’ comment: Thank you for highlighting this. We strongly agree and have corrected the sentence now stating a general need ‘for effective implementation strategies to increase PA levels in the primary years of childhood’
- Can you clarify whether MiPS is meant to run all year? Or is there a limited time during the school year that it runs?
Authors’ comment: Thank you for this comment. Please find that we have added that ‘the intervention was planned to run all year across all seasons’ (line 72-73).
Methods:
- Please identify your research philosophy (also called a paradigm or philosophical position)
Authors’ comment: Normally we do not add this information, as we have experienced that most reviewers found it redundant. Yet we believe that it is an important part of the scientific work when doing qualitative studies and helps strengthen the transparency and arguments of the study. Please find that we have added a paragraph stating the scientific theoretical foundation of the study and how it links to the aim (line 158-165).
- I’d like to see more consistency with the label of the ‘local leaders’ – they have so far been referred to as ‘local leaders’ ‘acting leaders’ and ‘acting local leaders’
Authors’ comment: Please find that this has been corrected. Only the terms ‘local leader’ or ‘preschool leader‘ is used throughout the text.
- We need to have more details about the interviewer: who conducted the interview? what was their relationship to the respondents (i.e., were they known to the interviewee? Would they have a reasonable level of trust with them to ensure full candor in their answers?) Might there be any biases related to this relationship? How experienced were the interviewer(s)?
Authors’ comment: Please find that we have added information on the interviewer in line 195-201.
- Please further describe the credibility of your data: was there any attempt to ensure that data was interpreted in a way that the interviewee intended? i.e. member validation either during or after the interview? Was data saturation reached when conducting your inductive analysis?
Authors’ comment: Thank you for this comment. Please find that we have addressed these concerns in line 225-226 (concerning saturation) and line 207-210 (concerning credibility).
Discussion:
- What degree of transference do you think exists in your data? How applicable might these results be to other Danish preschools? What about to an international population?
Authors’ comment: This is a relevant comment. Please find that we have added a new paragraph in the ‘Strengths and limitations’ section addressing the transparency and applicability of the results (line 582-594).

Reviewer 3 Report
The idea of the study is interesting, my recommendations are the following:
In the abstract, I recommend mentioning descriptions that represent MiPS.
Following the revision of this article, I recommend that it be mentioned in the title - case study.
Lines 386-388 recommend deleting the aim, as it is mentioned above.
Author Response
Authors’ comment: Thank you for your review of the manuscript. We are glad that you find the idea of the study interesting. We have strived to address your suggestions below. All the revisions are marked in yellow in the revised manuscript.
- In the abstract, I recommend mentioning descriptions that represent MiPS.
Authors’ comment: We agree and have added a short description in the abstract (line 13-15)
- Following the revision of this article, I recommend that it be mentioned in the title - case study.
Authors’ comment: Thank you for this comment. The title has now been adjusted to: The role of local leaders in the implementation of adult-initiated motor-skill and physical activity in preschool – a case study
- Lines 386-388 recommend deleting the aim, as it is mentioned above.
Authors’ comment: We have rephrased the first paragraph of the discussion. We have deleted the direct repetition of the aim and are instead arguing for the relevance of the study.
